# Prefrontal dopamine regulates fear reinstatement through the downregulation of extinction circuits

Natsuko Hitora-Imamura[1†], Yuki Miura[1], Chie Teshirogi[1], Yuji Ikegaya[1,2], Norio Matsuki[1], Hiroshi Nomura[1*‡]

[1]Laboratory of Chemical Pharmacology, Graduate School of Pharmaceutical Sciences, University of Tokyo, Tokyo, Japan; [2]Center for Information and Neural Networks, Osaka, Japan

**Abstract** Prevention of relapses is a major challenge in treating anxiety disorders. Fear reinstatement can cause relapse in spite of successful fear reduction through extinction-based exposure therapy. By utilising a contextual fear-conditioning task in mice, we found that reinstatement was accompanied by decreased c-Fos expression in the infralimbic cortex (IL) with reduction of synaptic input and enhanced c-Fos expression in the medial subdivision of the central nucleus of the amygdala (CeM). Moreover, we found that IL dopamine plays a key role in reinstatement. A reinstatement-inducing reminder shock induced c-Fos expression in the IL-projecting dopaminergic neurons in the ventral tegmental area, and the blocking of IL D1 signalling prevented reduction of synaptic input, CeM c-Fos expression, and fear reinstatement. These findings demonstrate that a dopamine-dependent inactivation of extinction circuits underlies fear reinstatement and may explain the comorbidity of substance use disorders and anxiety disorders.

*For correspondence: h-nomu@ umin.ac.jp

Present address: †Department of Pharmacology, Graduate School of Pharmaceutical Sciences, Hokkaido University, Sapporo, Japan; ‡Department of Psychiatry, University of North Carolina at Chapel Hill, Chapel Hill, United States

Competing interests: The authors declare that no competing interests exist.

## Introduction

Anxiety disorders are often treated with cognitive-behavioural interventions such as exposure therapy (*McNally, 2007*; *Vervliet et al., 2013*). Fear conditioning and extinction are used in animal models of anxiety disorders and their treatment (*Davis, 2002*). In extinction, conditioned responses can be reduced by prolonged presentations of conditional stimuli (CS) without the associated unconditional stimuli (US) (*LeDoux, 2000*). Many studies have shown that the infralimbic cortex (IL) is a critical brain region for extinction (*Herry et al., 2010*; *Sotres-Bayon and Quirk, 2010*). Extinction is suppressed by pharmacological inactivation of the IL (*Laurent and Westbrook, 2009*; *Sierra-Mercado et al., 2011*) as well as by local injection of N-methyl-D-aspartate receptor (NMDAR) antagonists (*Burgos-Robles et al., 2007*; *Sotres-Bayon et al., 2009*) or cannabinoid antagonists (*Lin et al., 2009*) into the IL. The IL inhibits the medial subdivision of the central nucleus of the amygdala (CeM), a key region for fear expression (*Wilensky et al., 2006*; *Ciocchi et al., 2010*), partly through intercalated amygdala neurons (ITCs) (*Likhtik et al., 2008*; *Amano et al., 2010*), which are also necessary for extinction.

Relapse is common in anxiety disorders. About 40% of patients in remission experience a relapse (*Bruce et al., 2005*; *Ansell et al., 2011*). While clinical observations have limitations on experimental control, relapse studies in the laboratory provide more information because of the greater potential for experimental manipulation (*Vervliet et al., 2013*). In experimental animals, fear can be reinstated by one or more US-only presentations after successful extinction (*Rescorla and Heth, 1975*; *Bertotto et al., 2006*). We previously reported that fear reinstatement occurs through NMDAR- and protein synthesis-dependent neural plasticity (*Shen et al., 2013*). It has also been reported that fear reinstatement requires β-adrenergic receptor activation, gamma-aminobutyric acid type A receptor

**eLife digest** Anxiety disorders affect millions of people worldwide. While many people with anxiety disorders can recover with appropriate treatment, about 40% of these individuals will encounter a relapse of their condition.

Researchers can investigate the causes of relapses by creating animal models of the processes involved. For example, if a mouse receives a small shock every time it enters a particular cage, it will learn to associate that cage with the shock. Once this association has been created, it can be 'undone' using a procedure called extinction. In the cage example, this may be performed by placing the mouse in the cage for a long time, but without giving it any shocks. Over time, the animal learns that the cage is no longer linked to an unpleasant outcome. However, if a mouse is given a reminder shock after extinction has occurred, the original association between the cage and the shock is re-established. This is known as fear reinstatement and is similar to a relapse.

A number of brain regions are thought to be involved in fear reinstatement. One such region, the amygdala, is heavily involved in fear responses. It is thought that another part of the brain, the medial prefrontal cortex (mPFC), can suppress the amygdala's responses, consequently reducing the animal's anxiety. While we have a good idea of which parts of the brain are involved in fear processing, we don't yet know how they work together to create a relapse.

Hitora-Imamura et al. used the aforementioned method of selectively giving mice small shocks when they entered cages to induce fear, extinction, and fear reinstatement and examined how this affected the mice's brain activity. As expected, fear could be linked to activity in the amygdala. During extinction, high levels of activity in the medial prefrontal cortex suppressed the amygdala's response. When the mice experienced the reminder shock, a chemical called dopamine was released. When dopamine entered the medial prefrontal cortex, the region's activity was reduced, removing the 'brakes' from the amygdala and reinstating the mice's fear.

The finding that dopamine is involved in fear reinstatement is particularly important, as many commonly abused drugs are known to increase levels of dopamine in the brain. Dopamine's role in fear reinstatement may explain why substance abuse is so closely linked to anxiety disorders.

endocytosis, and actin rearrangement in the basolateral amygdala (BLA) (*Lin et al., 2011*; *Motanis and Maroun, 2012*). However, neural circuit mechanisms responsible for fear reinstatement are poorly understood. Interestingly, the BLA–medial prefrontal cortex (mPFC) pathway is potentiated following fear extinction and depotentiated following reinstatement (*Vouimba and Maroun, 2011*). Therefore, the mPFC could be a key region mediating fear reinstatement. Nevertheless, little is known about activity changes of the mPFC and its downstream brain regions during fear reinstatement, synaptic modifications within the mPFC, and potential molecular regulation of such modifications.

To identify brain regions involved in processing fear reinstatement, we mapped the regional expression of the inducible immediate early gene (c-Fos). We focused on the mPFC, amygdala, and hippocampus because they are important in fear modulation and have reciprocal connections (*Quirk and Mueller, 2008*). In addition, we used in vitro patch-clamp recording to explore synaptic modifications within the mPFC. Building on our results from the c-Fos and electrophysiology experiments, we hypothesised that prefrontal dopamine plays a key role in reinstatement and tested this hypothesis pharmacologically. Together, these data suggest that a dopamine-dependent inactivation of extinction circuits underlies fear reinstatement.

## Results

### Reinstatement is associated with low c-Fos expression in the IL

To examine the neural circuits for fear reinstatement, we utilised a contextual fear-conditioning task, as described previously (*Shen et al., 2013*) (*Figure 1A*). Mice learned an association between CS (chamber A) and US (foot shocks) on Day 1. On Day 2, they received a prolonged CS presentation without any US (extinction training), then their freezing time gradually decreased. On Day 3, they were re-exposed to CS to confirm retention of extinction (test 1). To reinstate the conditioned fear, they immediately received a weak US (reminder shock) in chamber B on Day 4, and they were exposed to

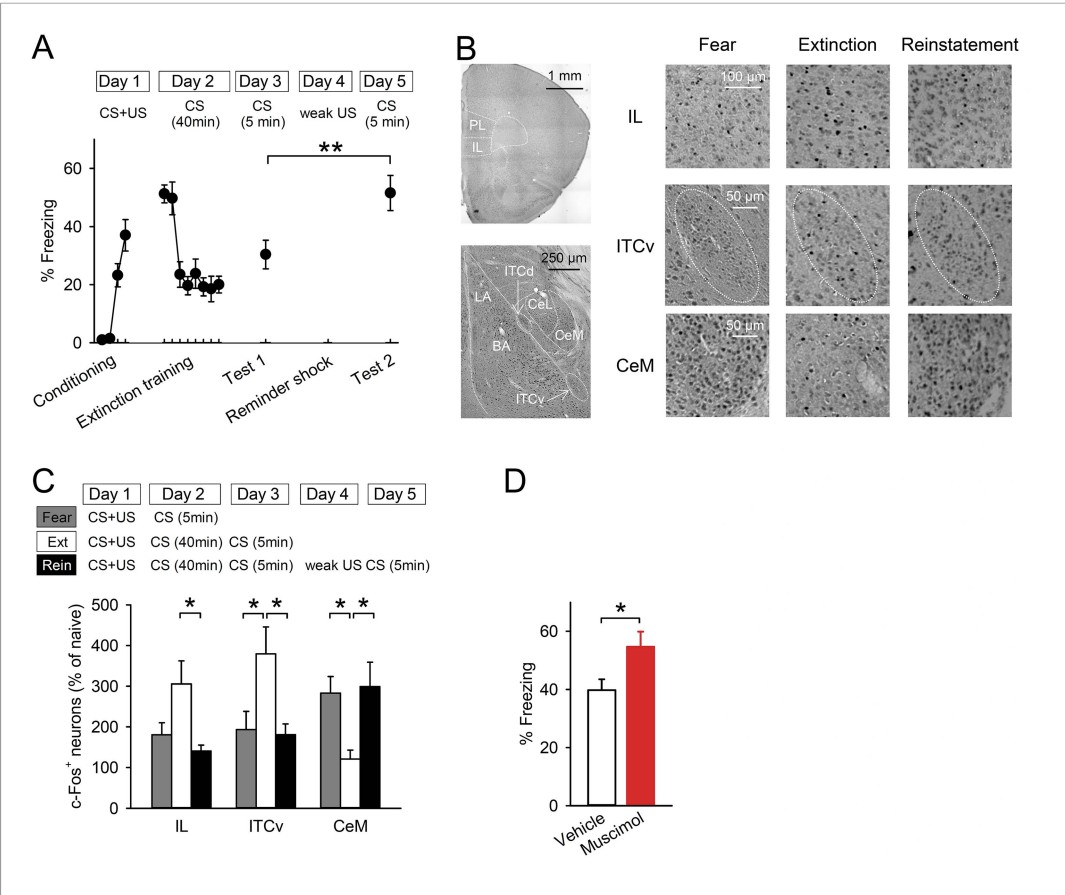

**Figure 1**. Reinstatement is associated with low IL activity. (**A**) A reminder shock reinstated extinguished fear (n = 10 mice; paired *t*-test, $t_{(9)}$ = 3.6, p = 0.0059). (**B**) Representative images of the infralimbic cortex (IL), the ventral intercalated amygdala neurons (ITCv), and the central nucleus of the amygdala (CeM) in the Fear, Extinction, and Reinstatement groups. (**C**) c-Fos+ cell density decreased in the IL and the ITCv and increased in the CeM with reinstatement (n = 8–11 mice; $F_{(2,27)}$ = 4.3, p = 0.023 [IL]; $F_{(2,26)}$ = 4.8, p = 0.0016 [ITCv]; $F_{(2,26)}$ = 6.3, p = 0.0058 [CeM]; Tukey's test, $P_{Extinction\ vs\ Reinstatement}$ = 0.029 [IL], 0.035 [ITCv], 0.013 [CeM]). (**D**) IL muscimol infusions resulted in high freezing (n = 10 mice; $t_{(18)}$ = 2.4, p = 0.030). **p < 0.01, *p < 0.05. Data represent mean ± standard error.

The following figure supplements are available for figure 1:

**Figure supplement 1**. Freezing behaviour of the mice subjected to c-Fos activity mapping.

**Figure supplement 2**. c-Fos+ cell density in the IL calculated in the analysis using a normal threshold and in an additional analysis using a strict threshold.

**Figure supplement 3**. The density of c-Fos+ cells in the PL, LA, BA, CeL, ITCd, CA1, CA2, CA3, and DG was comparable between the Extinction and Reinstatement groups (PL, $F_{(2,28)}$ = 3.6, p = 0.041; LA, $F_{(2,26)}$ = 1.4, p = 0.27; BA, $F_{(2,26)}$ = 0.068, p = 0.93; CeL, $F_{(2,26)}$ = 4.8, p = 0.017; ITCd, $F_{(2,27)}$ = 0.97, p = 0.39; CA1, $F_{(2,22)}$ = 1.3, p = 0.29; CA2, $F_{(2,22)}$ = 0.29, p = 0.75; CA3, $F_{(2,22)}$ = 1.0, p = 0.38; DG, $F_{(2,22)}$ = 0.46, p = 0.64; Tukey's test, CeL: $P_{Fear\ vs\ Extinction}$ = 0.015).

**Figure supplement 4**. Histological verification of cannula placements in the experiment with muscimol infusions into the IL.

CS again on Day 5 (test 2). The mice showed higher freezing time in test 2 than they did in test 1, suggesting successful reinstatement. Freezing time was comparable between tests 1 and 2 if mice were exposed to chamber B without a reminder shock on Day 4 (39.5 ± 4.2% in test 1 and 32.6 ± 5.3%

in test 2, n = 4). When we gave the reminder shock to naive mice, the reminder shock alone did not induce high fear responses (6.0 ± 2.1%, n = 5), indicating that the reminder shock-induced increase in freezing was derived from the original conditioned fear, not from new learning.

To identify the brain regions involved in processing reinstatement, we employed activity mapping with c-Fos immunohistochemistry. Mice were exposed to the CS one day after reminder shock (Reinstatement group). The Fear and Extinction groups were exposed to the CS one day after conditioning and one day after extinction training, respectively. The freezing time of the Reinstatement group was higher than it was in the Extinction group and comparable to that of the Fear group (*Figure 1—figure supplement 1*). Brains were removed 90 min later and subjected to c-Fos immunohistochemistry (*Figure 1B*). The density of c-Fos$^+$ cells in the IL in the Reinstatement group was lower than it was in the Extinction group and comparable to that in the Fear group (*Figure 1C*), which was not affected by thresholding (*Figure 1—figure supplement 2*). Given that the IL inhibits the CeM partly through the ITC, the reduced IL activity could result in low ITC and high CeM activities. Consistent with this idea, the density of c-Fos$^+$ cells in the ventral ITC and CeM decreased and increased, respectively, in the Reinstatement group compared to the Extinction group (*Figure 1C*). There were no significant differences between the Extinction and Reinstatement groups in other sub-regions of the mPFC, amygdala, or hippocampus (*Figure 1—figure supplement 3*). These results suggest that low IL activity disinhibits the CeM during fear reinstatement.

## Inactivation of the IL enhances fear responses

Next, we tested whether inactivation of the IL would lead to high fear responses. Mice underwent conditioning and extinction training. Muscimol, a gamma-aminobutyric acid type A receptor agonist, or a vehicle was infused into the IL 30 min before 5 min of re-exposure to the CS (*Figure 1—figure supplement 4*). Mice infused with muscimol showed higher freezing compared with those infused with a vehicle (*Figure 1D*), which is consistent with previous works in rats (*Quirk et al., 2000*; *Laurent and Westbrook, 2009*). These data suggest that inactivation of the IL is sufficient to enhance fear responses.

## Reinstatement is associated with presynaptic depression in the IL

To examine the cellular basis of lowered IL activity, we prepared brain slices 1 hr after the last test and obtained whole-cell recordings from pyramidal neurons in layer 5 of the IL. Frequency of miniature excitatory postsynaptic current (mEPSC) was lower in the Reinstatement group than it was in the Extinction group (*Figure 2A,B*), while mEPSC amplitude was comparable across groups (*Figure 2C*). Thus, excitatory synaptic inputs to the IL were decreased with reinstatement. To probe release probability, we measured paired-pulse ratio (PPR) by layer 2/3 stimulation. PPR was higher in the Reinstatement group than it was in the Extinction group (*Figure 2E,F*), indicating decreased transmitter release to IL neurons. Moreover, in the Reinstatement group, increased freezing time between tests 1 and 2 was negatively and positively correlated with mEPSC frequency and PPR, respectively (*Figure 2D,G*). These results suggest that presynaptic depression in the IL is associated with reinstatement.

To probe intrinsic neuronal excitability, the maximum number of action potentials generated during the current injections was also compared among the groups. The maximum number of action potentials in the Reinstatement group was not significantly different from either the Extinction group or the Fear group, while that of the Extinction group was higher than that of the Fear group, consistent with a previous study using auditory fear conditioning (*Santini et al., 2008*) (*Figure 2—figure supplement 1*). Other electrophysiological properties of IL neurons in the Reinstatement group were comparable to those in the Extinction and Fear groups (*Table 1*). Thus, intrinsic excitability of IL neurons did not change with fear reinstatement.

## A reminder shock activates dopaminergic ventral tegmental area neurons projecting to the IL

The mPFC, including the IL, receives dopaminergic innervation from the ventral tegmental area (VTA). It is reported that aversive stimuli activate VTA dopaminergic neurons (*Matsumoto and Hikosaka, 2009*; *Brischoux et al., 2009*) and elevate dopamine concentration in the PFC (*Abercrombie et al., 1989*; *Hamamura and Fibiger, 1993*). Additionally, dopamine application with electric stimulation

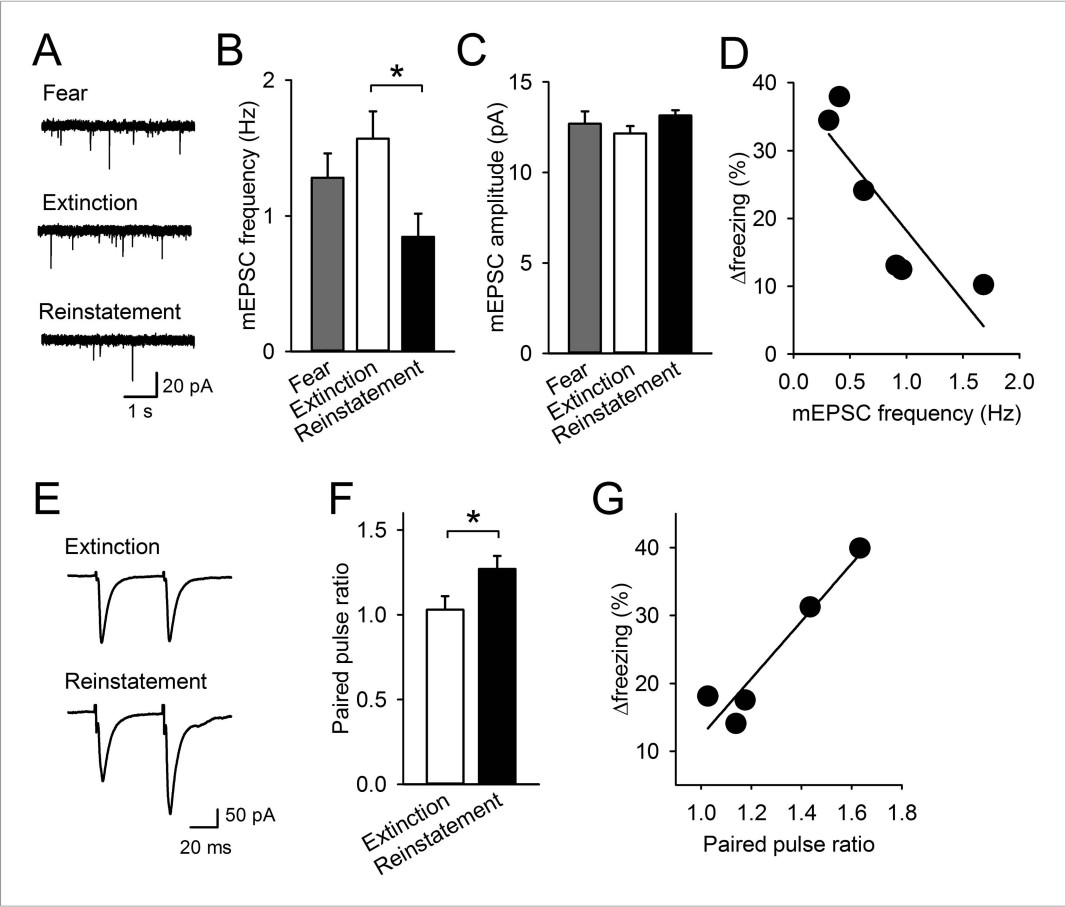

**Figure 2.** Reinstatement is associated with presynaptic depression in the IL. (**A**) Representative miniature excitatory postsynaptic current (mEPSC) traces. (**B**) IL neurons had lower mEPSC frequency in the Reinstatement group (n = 8 neurons from 6 mice) than the Extinction group (n = 8 neurons from 4 mice) ($F_{(2,21)}$ = 3.9, p = 0.037; $P_{Extinction\ vs\ Reinstatement}$ = 0.030). (**C**) mEPSC amplitude did not differ across groups ($F_{(2,21)}$ = 1.9, p = 0.38). (**D**) mEPSC frequency negatively correlated with Δfreezing (different degrees of freezing time between tests 1 and 2) in the Reinstatement group (r = −0.83, p = 0.040). (**E**) Representative traces of EPSCs evoked by paired-pulse stimulation. (**F**) IL neurons had a higher paired-pulse ratio (PPR) in the Reinstatement group (n = 8 neurons from 5 mice) than the Extinction group (n = 8 neurons from 4 mice) ($t_{(14)}$ = 2.2, p = 0.049). (**G**) PPR positively correlated with Δfreezing in the Reinstatement group (r = 0.95, p = 0.012). *p < 0.05. Data represent mean ± standard error.
The following figure supplement is available for figure 2:

**Figure supplement 1**. Intrinsic excitability of infralimbic neurons did not change with fear reinstatement.

suppresses transmitter release onto mPFC neurons via dopamine D1 receptors (D1Rs) (*Law-Tho et al., 1994*; *Gao et al., 2001*). Therefore, we hypothesised that a reminder shock activates the VTA-to-IL circuit and that dopamine D1 signalling in the IL contributes to reduction of synaptic input onto IL neurons and subsequent fear reinstatement. In order to assess this idea, we tested whether a reminder shock induces c-Fos expression in the VTA neurons projecting to the IL. We retrogradely labelled the neurons projecting to the IL by infusing Alexa 488-conjugated cholera toxin subunit B (CTB) into the IL (*Figure 3A*). Of the retrogradely labelled cells ($CTB_{IL}^+$) in the VTA, 59.1 ± 4.5% were immunopositive for a dopamine neuron marker, tyrosine hydroxylase ($TH^+$), indicating that they were dopaminergic. This is within the range reported in previous studies (*Margolis et al., 2006*; *Lammel et al., 2011*). The mice underwent conditioning, extinction training, test 1, and were exposed to chamber B with or without a reminder shock; their brains were removed 90 min later. c-Fos and TH were immunostained and observed in the VTA (*Figure 3B*). We found that a reminder shock increased

**Table 1**. Electrophysiological properties of IL neurons

|  | Fear | Extinction | Reinstatement |
|---|---|---|---|
| Resting potential (mV) | $-70.7 \pm 1.1$ | $-72.0 \pm 1.0$ | $-70.6 \pm 0.7$ |
| Input resistance (MΩ) | $276.5 \pm 24.6$ | $391.9 \pm 32.4$* | $362.6 \pm 30.9$ |
| Spike amplitude (mV) | $75.5 \pm 1.5$ | $72.9 \pm 2.0$ | $76.3 \pm 1.2$ |
| First interspike interval (ms) | $7.9 \pm 0.5$ | $8.8 \pm 0.6$ | $8.9 \pm 0.4$ |
| Rheobase (pA) | $78.1 \pm 5.7$ | $65.5 \pm 4.9$ | $70.0 \pm 6.1$ |
| Spike threshold (mV) | $-37.3 \pm 0.7$ | $-37.2 \pm 1.0$ | $-35.1 \pm 0.6$ |
| Voltage sag (mV) | $-3.0 \pm 0.3$ | $-3.1 \pm 0.2$ | $-3.8 \pm 0.4$ |
| Half width of spike (ms) | $1.01 \pm 0.03$ | $0.98 \pm 0.02$ | $1.04 \pm 0.02$ |
| fAHP (mV) | $-17.3 \pm 0.7$ | $-16.7 \pm 0.7$ | $-16.7 \pm 0.7$ |
| mAHP (mV) | $-1.6 \pm 0.5$ | $-0.9 \pm 0.4$ | $-1.0 \pm 0.5$ |

*$p < 0.05$ vs Fear, Tukey's test.
fAHP, fast afterhyperpolarization; mAHP, medium afterhyperpolarization.

c-Fos expression in the $CTB_{IL}^+$ $TH^+$ VTA neurons (*Figure 3C*), but not in the $CTB_{IL}^+$ $TH^-$ VTA neurons (*Figure 3D*). This result suggests that a reminder shock activates dopaminergic VTA neurons projecting to the IL.

## D1Rs in the IL mediate reinstatement

To test whether D1R signalling is involved in reinstatement, we infused a D1R antagonist, SCH23390, or vehicle 30 min before giving the mice a reminder shock and measured their freezing time in test 2 (*Figure 4—figure supplement 1*). SCH23390 infusions into the IL prevented reinstatement (*Figure 4A*). On the other hand, SCH23390 infusions into the prelimbic cortex (PL), a region adjacent

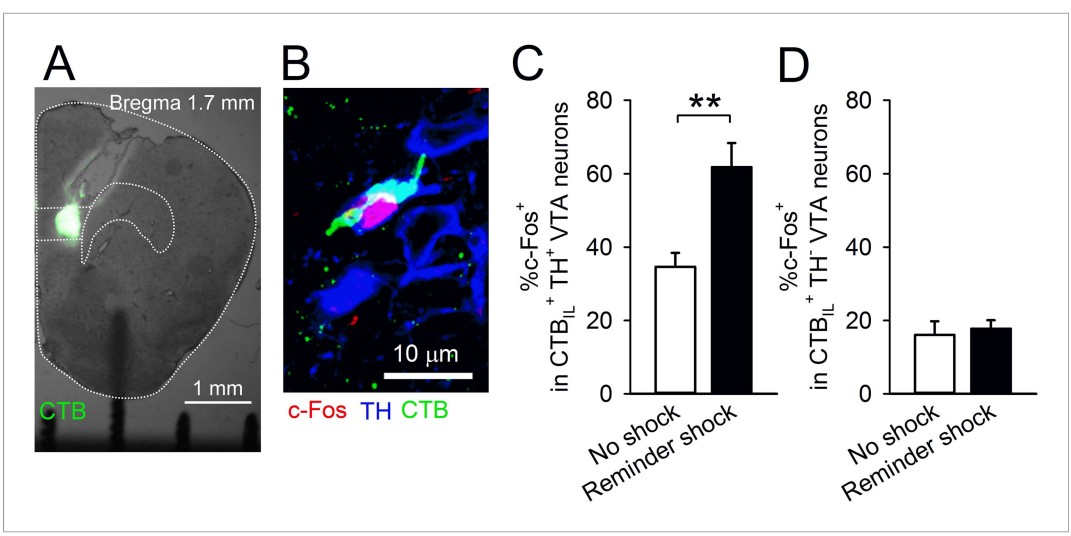

**Figure 3**. A reminder shock activates dopaminergic VTA neurons projecting to the IL. (**A**) Coronal brain section of a mouse with Alexa 488-conjugated cholera toxin subunit B (CTB) infusion into the IL. (**B**) A representative immunofluorescence image of the ventral tegmental area (VTA) neurons with c-Fos, tyrosine hydroxylase (TH), and CTB. (**C**) A reminder shock increased the proportion of c-Fos$^+$ neurons in IL-projecting $TH^+$ VTA neurons (no shock: n = 7, reminder shock: n = 6 mice; $t_{(11)}$ = 4.3, p = 0.0012). (**D**) A reminder shock did not increase the proportion of c-Fos$^+$ neurons in IL-projecting $TH^-$ VTA neurons (no shock: n = 7, reminder shock: n = 6 mice). **p < 0.01, Data represent mean ± standard error.

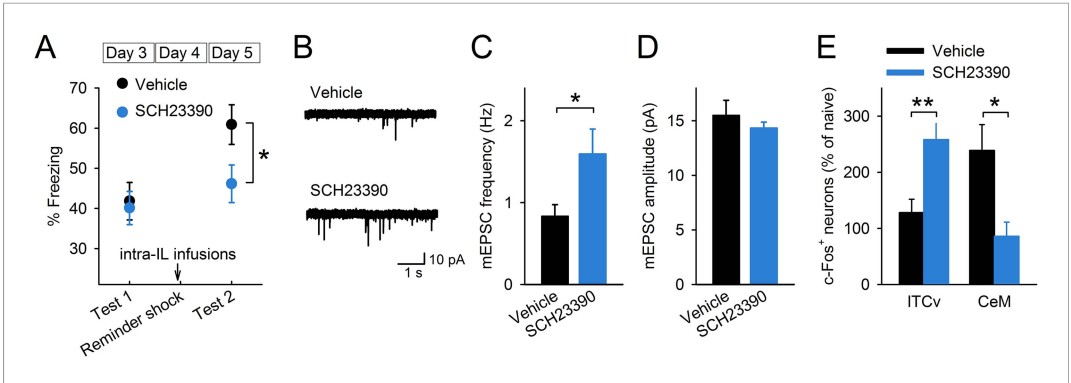

**Figure 4**. D1Rs in the IL mediate reinstatement. (**A**) SCH23390 infusions into the IL before a reminder shock suppressed reinstatement (phosphate-buffered saline [PBS]: n = 15, SCH23390: n = 14 mice; $t_{(27)}$ = 2.2, p = 0.039). (**B**) Representative mEPSC traces. (**C**) SCH23390-infused mice demonstrated higher mEPSC frequency (n = 9, 10 neurons; $t_{(17)}$ = 2.2, p = 0.044). (**D**) SCH23390 infusions had no effects on mEPSC amplitude. (**E**) SCH23390-infused mice demonstrated higher and lower c-Fos+ cell density in the ventral intercalated amygdala neurons (ITCv) and the central nucleus of the amygdala (CeM), respectively (n = 7–8 mice; ITCv: $t_{(13)}$ = 3.0, p = 0.0093; CeM: $t_{(14)}$ = 2.9, p = 0.011). **p < 0.01, *p < 0.05. Data represent mean ± standard error.

The following figure supplements are available for figure 4:

**Figure supplement 1**. Histological verification of cannula placements in the experiment with SCH23390 infusions into the infralimbic (**A**) and the prelimbic (**B**) cortices.

**Figure supplement 2**. SCH23390 infusions into the prelimbic cortex had no effects on reinstatement (n = 9 mice).

to the IL, did not affect reinstatement (*Figure 4—figure supplement 2*). These results indicate a specific role of IL dopaminergic signalling in the induction of reinstatement. Next, we examined the effect of IL D1R blockage on reduction of synaptic input onto IL neurons associated with reinstatement. Brain slices were prepared after test 2 from the mice infused with SCH23390 or vehicle into the IL before the reminder shock. mEPSC frequency was higher in the SCH23390-infused mice than it was in the vehicle-infused mice (*Figure 4B,C*), while mEPSC amplitude was comparable (*Figure 4D*). Thus, IL D1R blockage attenuated reduction of synaptic input associated with reinstatement. Finally, we examined the effect of IL D1R blockage on c-Fos expression in the amygdala during test 2. Brains were removed 90 min after test 2 from the mice infused with SCH23390 or vehicle into the IL before the reminder shock. SCH23390-infused mice showed higher and lower c-Fos expression than vehicle-infused mice in the ventral ITC and CeM, respectively (*Figure 4E*). Thus, we concluded that IL D1R blockage prevents c-Fos expression changes in the amygdala associated with reinstatement. Taken together, these results indicate that D1R signalling in the IL is necessary for reduction of synaptic inputs, CeM disinhibition, and reinstatement.

## Discussion

Prevention of relapse is a challenge in treating anxiety disorders. Fear reinstatement can cause relapse after exposure therapy. Accordingly, we investigated the neural circuit mechanism of fear reinstatement. We found that a reminder shock decreased IL and ventral ITC activity and increased CeM activity as indexed by c-Fos expression. Reinstatement was accompanied by presynaptic depression of transmitter release onto IL neurons. Moreover, we found that a reminder shock activated IL-projecting dopaminergic neurons in the VTA, and the blocking of IL D1R signalling prevented reduction of synaptic inputs, activity changes of ventral ITC and CeM, as well as fear reinstatement. These findings suggest that a dopamine-dependent inactivation of extinction circuits underlies fear reinstatement.

We compared c-Fos expression induced by an exposure to the experimental context before and after a reminder shock (Extinction group and Reinstatement group) in order to identify brain regions

involved in processing fear reinstatement. Activity mapping with c-Fos is a well-established and often used method to identify brain regions that are involved in processing motivation behaviour, social behaviour, learning and memory, and so on (*Zhang and Kelley, 2000*; *Tronel and Sara, 2002*; *Frankland et al., 2004*; *Veening et al., 2005*; *Makino et al., 2015*). Other studies reporting that neuronal activation patterns detected by c-Fos expression and by functional magnetic resonance imaging correlate well (*Lu et al., 2004*; *Lazovic et al., 2005*) also support the utility of c-Fos immunohistochemistry to determine changes in neuronal activity. It is important to note, however, that c-Fos expression is not the same as neuronal firing activity. Because c-Fos expression is dependent on an increase in $Ca^{2+}$ (*Lerea et al., 1992*), firing activity does not always result in c-Fos elevation (*Clayton, 2000*). Further studies using in vivo electrophysiological recordings (such as local field potential) are needed to directly reveal the neuronal firing activity during reinstatement.

We found a remarkable decrease in c-Fos expression followed by reinstatement in the IL but not in the PL (*Figure 1C*, *Figure 1—figure supplement 3*). Both the IL and PL of the mPFC have important roles in fear regulation. The PL promotes fear responses. As indicated by our data and previous reports, neuronal activity of the PL, which is elevated during fear retrieval, is not elevated (or lowered) during extinction retrieval (*Burgos-Robles et al., 2009*; *Sotres-Bayon et al., 2012*). On the other hand, the IL increases its activity during extinction retrieval (*Milad and Quirk, 2002*), suggesting the negative control of fear expression by the IL. In this study, we found that c-Fos expression in the Reinstatement group, compared with the Extinction group, was not changed in the PL, and it was lowered in the IL. Our findings suggest that fear reinstatement is regulated by downregulation of extinction circuits.

Low IL activity may cause ITC downregulation and CeM upregulation to reinstate fear. We found that reinstatement was accompanied by low IL and ventral ITC c-Fos expression (*Figure 1C*). Local injection of muscimol into the IL resulted in higher freezing (*Figure 1D*), indicating that IL inactivation is sufficient for reappearance of extinguished fear. The IL has direct projections to the ITC (*Cho et al., 2013*), which is critically involved in fear extinction (*Likhtik et al., 2008*; *Busti et al., 2011*; *Mańko et al., 2011*). The ITC provides feedforward inhibition of output neurons in the CeM (*Royer et al., 1999*). Therefore, low activity of the IL and ITC could result in disinhibition of the CeM. Accordingly, we found that reinstatement was accompanied by elevated c-Fos expression in the CeM (*Figure 1C*). The CeM projects to brain structures controlling conditioned fear responses, including the periaqueductal grey and the ventromedial and lateral hypothalamus (*Hopkins and Holstege, 1978*; *Veening et al., 1984*; *Cassell et al., 1986*), and CeM activity is necessary and sufficient for expression of freezing (*Wilensky et al., 2006*; *Ciocchi et al., 2010*). Thus, disinhibition of the CeM by downregulation of IL–ITC pathway may underlie fear reinstatement. Another possible pathway is through the lateral subdivision of the central nucleus of the amygdala (CeL), which also receives projections from the IL and contains two functionally distinct subpopulations of neurons ($CEl_{on}$ or somatostatin-expressing neurons, and $CEl_{off}$ or protein kinase C-δ-expressing neurons) forming local inhibitory circuits to inhibit CeM activity (*Ciocchi et al., 2010*; *Haubensak et al., 2010*; *Li et al., 2013*). Although we did not find a significant difference between the Extinction and Reinstatement groups in the CeL (*Figure 1—figure supplement 2*), further analysis with a distinction between those two populations might reveal the activation of $CEl_{on}$ neurons, which suppress $CEl_{off}$ neurons and lead to disinhibition of the CeM.

Our results provide a novel insight into the prefrontal dopaminergic modulation of an aversive memory. Many studies have shown that dopamine neurons are activated in response to appetitive stimuli, and that dopamine signalling affects reward-related behaviour and memory (*Schultz, 1997*). Although it has also been reported that aversive stimuli activate midbrain dopaminergic neurons (*Brischoux et al., 2009*; *Matsumoto and Hikosaka, 2009*), the exact role of dopamine released in aversive situations is poorly understood. We found that reinstatement-inducing stimuli elevated c-Fos expression in the dopaminergic VTA neurons projecting to the IL (*Figure 3C*). Moreover, blocking dopamine signalling in the IL prevented fear reinstatement (*Figure 4A*), suggesting the critical role of dopamine in reviving the aversive memory.

Further studies are required to identify how VTA dopaminergic neurons may be recruited by a reminder shock. Anatomically, VTA dopaminergic neurons that receive innervations from the lateral habenula preferentially project to the mPFC (*Lammel et al., 2012*). The habenula receives input from limbic system and this circuit is implicated in aversive information processing (*Hikosaka, 2010*; *Okamoto et al., 2012*). Thus, it is possible that a reminder shock activates the habenula–VTA circuit and subsequent dopaminergic signalling in the IL.

We found that reinstatement was accompanied by dopamine D1R-dependent reduction of synaptic input in the IL (*Figure 4B,C*). mEPSC frequency was low and paired-pulse ratio was high in the IL neurons of the Reinstatement group, and IL D1R blockage reversed the low mEPSC frequency. An additional experiment using minimal stimulation would be helpful to examine changes in release probability upon fear reinstatement. Previous in vitro studies also revealed the inhibitory effects of prefrontal D1Rs by pharmacological and genetic manipulations. Dopamine attenuates excitatory synaptic transmission in prefrontal neurons in a D1R-dependent manner (*Gao et al., 2001*; *Mair and Kauer, 2007*). Application of dopamine, combined with electrical stimulation, can induce long-term synaptic depression in the mPFC (*Law-Tho et al., 1995*; *Huang et al., 2004*); however, the phenomenon is not observed in heterozygous D1R knockout mice (*Huang et al., 2004*). The signalling mechanisms underlying dopamine-mediated presynaptic depression remain to be determined. One possible mechanism is that dopamine triggers adenosine release and subsequently activates presynaptic adenosine A1 receptors. D1Rs and adenosine-mediated presynaptic depression have also been reported in the VTA of guinea pigs (*Bonci and Williams, 1996*) and in the basal ganglia of zebra finches (*Ding et al., 2003*).

The present findings suggest a possible dopaminergic mechanism of fear reinstatement as follows: a reminder shock activates VTA dopaminergic neurons projecting to the IL, and dopamine D1R signalling lowers IL activity with presynaptic depression, which could result in low activity of the ventral ITC, thereby disinhibiting CeM and reinstating a once extinguished fear. Previous studies have shown that drugs of abuse increase dopamine release in both animals (*Di Chiara and Imperato, 1988*; *Chen et al., 1990*) and humans (*Laruelle et al., 1995*; *Volkow et al., 1999*; *Drevets et al., 2001*). This dopaminergic mechanism of reinstatement may explain the high rate of comorbid substance use disorders with anxiety disorders (*Kendler et al., 1996*).

## Materials and methods

### Animals

All experiments were approved by the animal experiment ethics committee at The University of Tokyo (approval number 24-10) and were in accordance with The University of Tokyo guidelines for the care and use of laboratory animals. Male C57BL/6J mice (8–15 weeks old; SLC, Shizuoka, Japan) were housed in group cages of four under standard laboratory conditions (12-hr light/12-hr dark cycle, with light from 7 a.m. to 7 p.m. and free access to food and water). Mice were handled daily for 1 week and housed individually for 2 days before behavioural procedures. All behavioural procedures were performed between 9 a.m. and 2 p.m.

### Contextual fear conditioning

Behavioural procedures for fear conditioning, extinction, and reinstatement were performed in accordance with our previous protocol (*Shen et al., 2013*). For contextual fear conditioning, after a 150-s acclimation period in transparent rectangular conditioning chamber A (18 cm wide, 15 cm deep, 27 cm high) with white light and a stainless steel grid floor, 3 shocks (1 mA, 2 s) were delivered through a shock scrambler (SGS-003DX; Muromachi Kikai, Tokyo, Japan) with a 150-s interval between shocks. Mice were left in the chamber for an additional 60 s and then returned to their home cages. The entire duration of this session was 510 s. For extinction training and testing, mice were placed in chamber A without any shocks for 40 min and 5 min, respectively. A reminder shock (0.6 mA, 2 s) was given immediately after the mice were placed in white triangular chamber B (22 cm wide, 19 cm deep, 27 cm high) with red light and a stainless steel grid floor. The mice then returned to their home cages. Unless otherwise mentioned, testing sessions were 5 min.

Mice in the Fear group underwent contextual fear conditioning on Day 1 and testing on Day 2. Mice in the Extinction group underwent contextual fear conditioning on Day 1, extinction training on Day 2, and testing on Day 3. Mice in the Extinction group of the PPR experiment underwent contextual fear conditioning on Day 1, extinction training on Day 2, testing on Day 3, exposure to chamber B on Day 4, and testing in chamber A on Day 5. Mice in the Reinstatement group underwent contextual fear conditioning on Day 1, extinction training on Day 2, testing on Day 3, a reminder shock on Day 4, and testing on Day 5. Each session was video recorded for automatic scoring of freezing according to a previously described method (*Nomura and Matsuki, 2008*). Freezing was defined as the absence of all movement except those related to breathing. Naive mice were kept in their home

cages and were not exposed to the conditioning apparatus. The numbers of mice used in the c-Fos immunohistochemistry are as follows: (naive, Fear, Extinction, Reinstatement) = (9, 11, 11, 9) in PL and IL; (8, 10, 11, 8) in the lateral and basal nuclei of the amygdala, CeM and CeL; (8, 11, 11, 8) in dorsal ITC; (8, 11, 10, 8) in ventral ITC; and (8, 8, 9, 8) in CA1, CA2, CA3, and dentate gyrus.

## Immunohistochemistry and microscopy

Mice were perfused intracardially with phosphate-buffered saline (PBS) followed by 4% paraformaldehyde 90 min after behavioural tests. Brains were removed and stored in the same fixative for 8 hr at 4°C and subsequently immersed in 20% and 30% sucrose for 24 hr and 48 hr at 4°C. The immunocytochemical staining was performed on 40-μm thick free-floating sections that were prepared using a cryostat (HM520; Thermo Fisher Scientific, Waltham, MA, USA).

For c-Fos staining with diaminobenzidine (DAB), the sections were incubated in 0.2% Triton-X-100 for 15 min and 0.03% $H_2O_2$ for 30 min. The sections were incubated with a polyclonal anti-c-Fos antibody (Anti-c-Fos (Ab-5) (4–17) rabbit, 1:5000, Calbiochem, San Diego, CA, USA) for 48 hr at 4°C, goat anti-rabbit biotinylated secondary antibody (BA-1000, 1:500; Vector Laboratories, Burlingame, CA, USA) for 2 hr, VECTASTAIN ABC Kit (Vector Laboratories) for 1.5 hr, and DAB solution (349-00903, 0.03%, Wako, Osaka, Japan) with 0.01% $H_2O_2$ for 7–10 min. The sections were mounted on slides, air-dried, dehydrated in ethanol solutions and xylene, and cover slipped with marinol. Images of the mPFC (bregma 2.2 to 1.5 mm), amygdala (bregma −1.2 to −1.8 mm), and hippocampus (bregma −1.5 to −2.0 mm) were acquired using a microscope (Leica AF6000, 10× objective lens [NA, 0.3], Leica, Germany). All cell counting experiments were conducted blind to experimental group. The quantification of c-Fos-positive cells was performed with ImageJ software (Scion, Frederick, MD, United States). c-Fos immunoreactive cells were counted bilaterally using at least three sections for each area. Sub-regions of the mPFC, amygdala, and hippocampus were outlined as a region of interest (ROI) according to the Paxinos and Franklin atlas. c-Fos-positive nuclei were counted relative to a counting threshold based on staining intensity and target size. The parameters of the counting threshold were set based on a standard control slide from each staining run. The mean density in each structure for each animal was divided by the mean density in that region for the naive control group in order to generate a normalized density for each animal. These normalized data were expressed as a percentage, and these percentages were averaged across mice in order to produce the mean of each group.

For fluorescence immunohistochemistry, the sections were incubated with primary antibodies, including a polyclonal anti-c-Fos antibody (1:1000) and mouse anti-tyrosine hydroxylase antibody (MAB318, 1:500; Millipore, MA, United States), for 24 hr at 4°C, and secondary antibodies, including a goat anti-rabbit biotinylated antibody (BA-1000, 1:500; Vector Laboratories) and Alexa Fluor 405 goat anti-mouse IgG secondary antibody (A31553, 1:400; Life Technologies, CA, United States) for 2 hr, VECTASTAIN ABC Kit (Vector Laboratories) for 1.5 hr, and TSA-Cyanine 3 (SAT704A001EA, 1:1000; Perkin–Elmer, Waltham, MA, USA) for 1 hr. The sections were mounted in PermaFluor (ThermoShandon, Pittsburgh, PA, United States). Images of the VTA (bregma −2.9 to −3.4 mm) were acquired using a confocal microscope (CV1000, 40× objective lens (NA, 1.3); Yokogawa, Tokyo, Japan). All cell counting experiments were conducted blind to experimental group. The quantification of c-Fos-positive cells was performed with ImageJ software (Scion). CTB positive and TH and c-Fos immunoreactive cells were counted bilaterally using at least five sections (374 cells from 13 mice). The VTA were outlined as an ROI according to the Paxinos and Franklin atlas. The number of c-Fos-positive cells in the CTB[+] and TH[+] cells was calculated by thresholding c-Fos immunoreactivity above background levels. The percentage for each animal was averaged across mice in order to produce the mean of each group.

## Surgery

Under intraperitoneal xylazine (10 mg/kg) and pentobarbital (2.5 mg/kg) anaesthesia, 26-gauge stainless steel guide cannulas (Plastics One, Roanoke, VA, United States) were implanted aimed at the IL (A/P 1.7 mm, L/M ±0.3 mm, D/V −3.0 mm) or the PL (A/P 2.0 mm, L/M ±0.3 mm, D/V −2.2 mm). These cannulas were secured to the skull using a mixture of acrylic and dental cement, and 33-gauge dummy cannulas were then inserted into each guide cannula to prevent clogging. Mice were given at least 7 days of postoperative recovery time.

## Drugs and microinfusions

Mice underwent fear conditioning on Day 1, extinction training on Day 2, testing on Day 3, and received bilateral infusions of PBS or muscimol (0.25 μg/side) into the IL 30 min before testing on Day 4.

Mice underwent fear conditioning on Day 1, extinction training on Day 2, testing on Day 3, received bilateral infusions of PBS or SCH23390 (1 μg/side) into the IL or PL 30 min before the reminder shock on Day 4, and testing on Day 5. The numbers of mice used in the c-Fos immunohistochemistry are as follows: (PBS, SCH23390) = (8, 8) in CeM and (7, 8) in ventral ITC.

Alexa 488-conjugated CTB (0.5 μg/side, Life Technologies) was infused into the IL 3 days before the beginning of behavioural procedures. Mice underwent fear conditioning on Day 1, extinction training on Day 2, testing on Day 3, and exposed to the chamber B with or without reminder shock on Day 4 (reminder shock group and no shock group, respectively). Brains were removed 90 min later.

Infusions were made over 2 min, and the infusion cannulas (28 gauge, extending 0.5 mm below the guide cannula) were left in place for at least 1 min afterwards.

## Electrophysiology

Mice were deeply anaesthetised with diethyl ether and decapitated 60–90 min after re-exposure to the conditioning context. Brains were removed quickly, and 300-μm thick coronal slices containing the IL were prepared with a vibratome (VT 1200S, Leica) in ice-cold, oxygenated (95% $O_2$/5% $CO_2$) modified artificial cerebrospinal fluid containing 222.1 mM sucrose, 27 mM $NaHCO_3$, 1.4 mM $NaH_2PO_4$, 2.5 mM KCl, 0.5 mM ascorbic acid, 1 mM $CaCl_2$, and 7 mM $MgSO_4$.

Picrotoxin (100 μM) was added to ACSF (artificial cerebrospinal fluid) (127 mM NaCl, 1.6 mM KCl, 1.24 mM $KH_2PO_4$, 1.3 mM $MgSO_4$, 2.4 mM $CaCl_2$, 26 mM $NaHCO_3$, 10 mM glucose) to block inhibitory synaptic currents. Whole-cell patch-clamp recordings were performed with glass microelectrodes (3–6 MΩ) filled with internal solution (120 K-gluconate, 5 KCl, 10 4-(2-hydroxyethyl)-1-piperazineethanesulfonic acid, 1 $MgCl_2$, 10 phosphocreatine-$Na_2$, 2 MgATP, 0.1 $Na_2$GTP, 0.2 ethylene glycol tetraacetic acid, pH 7.2–7.3, 280–295 mOsm). For electrical stimulation, a pipette with a large tip (∼3 μm) was filled with ACSF and placed in layer 2/3. Brief current pulses (50 μs, 1–40 μA) were delivered with a stimulation-isolation unit (Nihon Kohden, Tokyo, Japan). Paired stimuli were given with an interstimulus interval of 50 ms, and the ratio between the amplitude of the second and first EPSCs was calculated. mEPSCs were recorded at a holding potential of −70 mV in the presence of tetrodotoxin (TTX, 1 μM). mEPSCs were detected using an in-house MATLAB programme and were defined as inward currents with amplitudes greater than 7 pA (*Miura et al., 2012*). To examine the intrinsic excitability, IL neurons were injected with 800-ms depolarizing current pulses ranging from 40 pA to 400 pA at 40-pA increments. The number of action potentials evoked by each current intensity was counted. The amplitude of fast afterhyperpolarization was calculated as the difference between the minimum potential after the second evoked spike within the 800-ms pulse and the spike threshold. To measure the medium afterhyperpolarization (mAHP), cells were held at −70 mV, and the 800-ms pulse, which evoked two action potentials, was injected. The amplitude of mAHP was calculated as the difference between the negative peak of the potential after the end of the 800-ms pulse and the resting membrane potential (−70 mV). To measure the voltage sag, a hyperpolarizing current pulse of 200 pA was injected in current-clamp mode. The voltage sag was calculated by subtracting the average steady-state voltage during a 100-ms period beginning 645 ms after the beginning of the hyperpolarizing step minus the peak of the hyperpolarization. Input resistance was calculated by current response to a 10-mV, 30-ms depolarizing pulse in voltage-clamp mode. Data were sampled at 20 kHz and filtered at 2 kHz using an Axopatch 700B amplifier (Axon Instruments, Foster, CA, United States), Digidata 1440A (Axon Instruments), and pClamp 10.2 (Molecular Devices, Sunnyvale, CA, United States). All data were acquired, stored, and analysed using Clampex 10, Clampfit, and MATLAB.

## Data analysis

All values are reported as mean ±SEM. Repeated measures analysis of variance (ANOVA), Tukey's test after one-way ANOVA, Student's *t*-tests, and paired *t*-tests were performed to identify significant differences.

## Acknowledgements

We thank the University of Tokyo/Leica microsystems imaging center for obtaining the imaging data. This work was supported by a Grant-in-Aid for JSPS Fellows (12J09784, to NHI), a Grant-in-Aid for Young Scientists (B) (25830002, to HN), and Grants-in-Aid for Scientific Research on Innovative Areas, 'Mesoscopic Neurocircuitry' (No. 23115101, to HN), 'The Science of Mental Time' (No. 26119507 to HN and No. 25119004 to YI), and 'Memory Dynamism' (No. 26115509 to HN).

## Additional information

### Funding

| Funder | Grant reference | Author |
|---|---|---|
| Japan Society for the Promotion of Science (JSPS) | 12J09784 | Natsuko Hitora-Imamura |
| Japan Society for the Promotion of Science (JSPS) | 25830002 | Hiroshi Nomura |
| Ministry of Education, Culture, Sports, Science, and Technology (MEXT) | 23115101 | Hiroshi Nomura |
| Ministry of Education, Culture, Sports, Science, and Technology (MEXT) | 26119507 | Hiroshi Nomura |
| Ministry of Education, Culture, Sports, Science, and Technology (MEXT) | 26115509 | Hiroshi Nomura |
| Ministry of Education, Culture, Sports, Science, and Technology (MEXT) | 25119004 | Yuji Ikegaya |

The funders had no role in study design, data collection and interpretation, or the decision to submit the work for publication.

### Author contributions

NH-I, Conception and design, Acquisition of data, Analysis and interpretation of data, Drafting or revising the article; YM, Acquisition of data, Analysis and interpretation of data, Drafting or revising the article; CT, Acquisition of data, Drafting or revising the article; YI, NM, Conception and design, Drafting or revising the article; HN, Conception and design, Analysis and interpretation of data, Drafting or revising the article

### Author ORCIDs

Hiroshi Nomura, http://orcid.org/0000-0002-6172-4788

### Ethics

Animal experimentation: All experiments were approved by the animal experiment ethics committee at The University of Tokyo (approval number 24-10) and were in accordance with The University of Tokyo guidelines for the care and use of laboratory animals. All surgery was performed under xylazine and pentobarbital anesthesia, and every effort was made to minimize suffering.

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
