## [Decision Letter]

Thank you for submitting your work entitled “Prefrontal dopamine regulates fear reinstatement through the downregulation of extinction circuits” for peer review at *eLife.* Your submission has been favorably evaluated by Timothy Behrens (Senior editor) and two reviewers, one of whom is a member of the Board of Reviewing Editors.

The reviewers have discussed the reviews with one another and the Reviewing editor has drafted this decision to help you prepare a revised submission.

The reviewers were very enthusiastic about your novel findings on the role of the infralimibic cortex in fear reinstatement. Indeed, both felt that this study contains some highly novel findings that should be published. However, they also felt that additional data analysis is needed and/or a better explanation on the methods used, to clarify how c-Fos cells were quantified in the infralimbic area (see individual responses for details below). Images on the quality of c-Fos labelling are required as well as information on the success of retrograde tracing of dopaminergic fibers. Reviewer 1 was questioning how reliable c-Fos labelling can be correlated with discharge activity and therefore asked for LFP recordings in the infralimbic cortex. If electrophysiological data are available then we would propose to include them in the existing manuscript. However, if they are not present than the authors should mention in the Discussion section of the manuscript the limits of the interpretation of c-Fos labelling data.

*Reviewer #1*:

The manuscript by Hitora-Imamura et al examines the neuronal mechanisms underlying the reinstatement of fear by a combination of behavioral studies with c-Fos labeling and ex vivo electrophysiological recordings. They find that upon reinstatement of fear c-Fos labeling is reduced in the infralimbic cortex (IL) and the connected ITC but increased in the central nucleus of the amygdala (CEM), which is a key nucleus of fear expression. Ex vivo investigations show further that this behavior is accompanied by a reduced mEPSC frequency in Pyramidal cells (PCs) recorded in the IL and paired pulse depression of evoked EPSCs indicating. The authors argue that presynaptic depression in release probability may cause the reduced recruitment of IL pyramidal cells and thus underlie the observed reinstatement of fear. Finally, they find that c-Fos labeling is enhanced in the VTA, specifically in cells projecting to the IL as revealed by retrograde tracers, and show that infusion of a D1 receptor antagonist into IL reduces freezing in reinstatement test. Moreover, they observe in ex vivo whole cell recordings an enhanced mEPSC frequency and c-Fos labelling in PCs of the IL. Thus blocking D1 receptors may be an important mechanism for reinstating freezing.

1) The entire study is based on the interpretation of c-Fos labeling being enhanced upon enhanced discharge activity. Since c-Fos expression is Ca^2+^ dependent, I do have my doubts whether this is always the case and would like to see at least some electrophysiological evidence (if available) such as local field potential (LFP) recordings in the IL and VTA or CEM to support this interpretation. If true than we should see a reduced power of the LFP signal in the IL upon reinstatement and enhanced LFP power in the VTA and CEM. This experiment would be very helpful (if available) to support the interpretation of the presented c-Fos data. Alternatively, discuss possible difficulties in the correlation between c-Fos labeling and neuronal discharge activity.

2) I did not find information in the Materials and Methods section on how the % change in c-Fos labeled cells was quantified. This is critical because the results and the combined interpretations depend on this analysis. Did the authors count c-Fos positive cells in a given area of brain regions? What is the number of cells and slices in which cells were counted? A double blind approach would also be important. Labeling intensity can vary among slices as well as animals. Which thresholds have been defined and how are the data dependent on thresholding? Moreover, it would be important to show c-Fos labeling in the three brain regions in the manuscript (specifically in Figure 1).

3) Figure 2: I am surprised about the low number of recorded cells. Figure 2 is based on 8 recordings from 6 mice (1.3 recordings per mouse) for the reinstatement group and 8 neurons from 4 mice (2 recordings per mouse) in the control group. Do we indeed see in D and G only 6 and 5 data points, respectively? The data are highly significant considering the low number of recorded cells and moderate number of mice. However, in order to support the finding on changes in release probability upon reinstatement, further data on failures of transmission (1st EPSC vs 2nd EPSC) and the size of the 1st EPSC (minimal stimulation; control vs reinstatement group) would be helpful. Moreover, why did the authors not test the paired pulse ratio in Figure 4 upon blocking D1 receptors?

4) It remains unclear how VTA cells may be recruited during reinstatement of fear. Direct connections exist between the IL and the VTA but also other connections from the habenula may play a role. Thus, the Discussion should include a chapter in which possible circuit mechanisms in VTA recruitment during reinstatement should be discussed.

5) During fear, extinction activity in the prefrontal cortex is reduced. In a recent publication by Sotres-Bayon et al. (Neuron 2012) it was shown that upon fear, extinction activity of PCs in the medial prefrontal activity was reduced. It would be important to discuss whether the authors believe that these are the same or different PC populations controlling extinction and reinstatement of fear or different populations. This is therefore critical because the Title of the presented study states that ‘reinstatement of fear is regulated by down regulation of fear extinction circuits’ and proposes that this is the same circuitry. This was unclear from the Discussion.

Reviewer #2:

Overall, this is a nice set of experiments exploring the role of the infralimbic cortex in fear reinstatement. The integration of behavior, electrophysiology, and immunohistological data provide convincing evidence that IL suppression through D1 activity is necessary for reinstatement.

1) My primary concern is that there is insufficient methodological detail for the quantification of c-Fos^+^ cells. How many sections per animal were used? Was any kind of unbiased stereology used to identify brain structure borders? Were cell counts averaged per animal? The numbers are expressed in the figures as % of naïve animals, but it is impossible to know what these numbers really mean without more information. Moreover, no micrographs are shown.

2) Additionally, it would be useful to know how successful the retrograde tracer was in labeling dopaminergic projections to the IL. Setting aside the question of c-Fos activation, what % of CTB^+^ cells were also TH^+fM^?

---

## [Author Response]

Reviewer#1:

*1) The entire study is based on the interpretation of c-Fos labeling being enhanced upon enhanced discharge activity. Since c-Fos expression is Ca*^*2+*^
*dependent, I do have my doubts whether this is always the case and would like to see at least some electrophysiological evidence (if available) such as local field potential (LFP) recordings in the IL and VTA or CEM to support this interpretation. If true than we should see a reduced power of the LFP signal in the IL upon reinstatement and enhanced LFP power in the VTA and CEM. This experiment would be very helpful (if available) to support the interpretation of the presented c-Fos data. Alternatively, discuss possible difficulties in the correlation between c-Fos labeling and neuronal discharge activity.*

Though c-Fos activity mapping is a common and useful method to identify brain regions involved in information processing, we agree with the reviewer about limitation of c-Fos mapping for detecting neuronal discharge activity and usefulness of electrophysiological experiment to support the c-Fos data. In accordance with the reviewer’s wishes, we have now added a paragraph in the Discussion section about technical limitation of c-Fos mapping as follows: “We compared c-Fos expression induced by an exposure to the experimental context before and after a reminder shock (Extinction group and Reinstatement group) in order to identify brain regions involved in processing fear reinstatement [...] Further studies using in vivo electrophysiological recordings (such as local field potential) are needed to directly reveal the neuronal firing activity during reinstatement.”. Also, we replaced the word “activation” and “activity” with “c-Fos expression” in the revised results.

*2) I did not find information in the Materials and Methods section on how the % change in c-Fos labeled cells was quantified. This is critical because the results and the combined interpretations depend on this analysis. Did the authors count c-Fos positive cells in a given area of brain regions? What is the number of cells and slices in which cells were counted? A double blind approach would also be important. Labeling intensity can vary among slices as well as animals. Which thresholds have been defined and how are the data dependent on thresholding? Moreover, it would be important to show c-Fos labeling in the three brain regions in the manuscript (specifically in*
Figure 1*).*

We have added an explanation for c-Fos labeling to the Material and Method section (the subsection ‘Immunohistochemistry and microscopy’) as follows: “All cell counting experiments were conducted blind to experimental group. The quantification of c-Fos-positive cells was performed with ImageJ software (Scion, Frederick, MD). c-Fos immunoreactive cells were counted bilaterally using at least three sections for each area […] These normalized data were expressed as a percentage, and these percentages were averaged across mice in order to produce the mean of each group.” Additionally, we analyzed our data with a strict threshold and got similar pattern of c-Fos^+^ expression as our previous analysis shown in the Figure 1 (shown in Figure 1—figure supplement 2).

Also, we added representative images of c-Fos labeling in the IL, ITC and CeM in Figure 1.

*3)*
Figure 2*: I am surprised about the low number of recorded cells.*
Figure 2
*is based on 8 recordings from 6 mice (1.3 recordings per mouse) for the reinstatement group and 8 neurons from 4 mice (2 recordings per mouse) in the control group. Do we indeed see in D and G only 6 and 5 data points, respectively? The data are highly significant considering the low number of recorded cells and moderate number of mice. However, in order to support the finding on changes in release probability upon reinstatement, further data on failures of transmission (1st EPSC vs 2nd EPSC) and the size of the 1st EPSC (minimal stimulation; control vs reinstatement group) would be helpful. Moreover, why did the authors not test the paired pulse ratio in*
Figure 4
*upon blocking D1 receptors?*

The experimental procedure and freezing behavior of the Vehicle group in Figure 4 were almost comparable to those of the reinstatement group in Figure 2. Because the mEPSC frequency of the Vehicle group in Figure 4 was almost same as that of the reinstatement group in Figure 2, the low mEPSC frequency by fear reinstatement is highly reproducible. But, we added the discussion about minimal stimulation in the revised Discussion.

The numbers of data points in Figure 2 are 6 and 5, respectively.

The purpose of the experiment for Figure 4 is to test whether activation of D1 receptors is involved in fear reinstatement and its associated changes. Because testing whether fear reinstatement is associated with presynaptic depression was done in Figure 2, the detailed analysis measuring the paired pulse ratio was not performed in Figure 4. But, we toned down the conclusion of Figure 4 and described that D1R signaling in the IL is necessary for “reduction of synaptic inputs”, which was originally “presynaptic depression”.

4) It remains unclear how VTA cells may be recruited during reinstatement of fear. Direct connections exists between the IL and the VTA but also other connections from the habenula may play a role. Thus, the Discussion should include a chapter in which possible circuit mechanisms in VTA recruitment during reinstatement should be discussed.

We added a Discussion chapter about possible circuit mechanisms in VTA recruitment by a reminder shock as follows: “Further studies are required to identify how VTA dopaminergic neurons may be recruited by a reminder shock. Anatomically, VTA dopaminergic neurons that receive innervations from the lateral habenula preferentially project to the mPFC (30). The habenula receives input from limbic system and this circuit is implicated in aversive information processing (52, 25). Thus, it is possible that a reminder shock activates the habenula–VTA circuit and subsequent dopaminergic signalling in the IL.”

5) During fear, extinction activity in the prefrontal cortex is reduced. In a recent publication by Sotres-Bayon et al. (Neuron 2012) it was shown that upon fear, extinction activity of PCs in the medial prefrontal activity was reduced. It would be important to discuss whether the authors believe that these are the same or different PC populations controlling extinction and reinstatement of fear or different populations. This is therefore critical because the Title of the presented study states that ‘reinstatement of fear is regulated by down regulation of fear extinction circuits’ and proposes that this is the same circuitry. This was unclear from the Discussion.

According to the previous studies including Sotres-Bayon et al., the activity of prelimbic neurons in the prefrontal cortex (PFC) decreases and the activity of infralimbic (IL) neurons in the PFC increases during fear extinction. Then, the IL has been proposed as a critical circuit for fear extinction. In the current study, we found that the decreased activity of IL neurons is associated with fear reinstatement. Therefore, we propose that fear reinstatement is regulated by down regulation of extinction circuits (that is the IL). We added to the Discussion as follows: “Low IL activity may cause ITC downregulation and CeM upregulation to reinstate fear […] The CeM projects to brain structures controlling conditioned fear responses, including the periaqueductal grey and the ventromedial and lateral hypothalamus (26, 11, 66), and CeM activity is necessary and sufficient for expression of freezing (14, 70). Thus, disinhibition of the CeM by downregulation of IL–ITC pathway may underlie fear reinstatement.”

Reviewer #2:

*1) My primary concern is that there is insufficient methodological detail for the quantification of c-Fos*^*+*^
*cells. How many sections per animal were used? Was any kind of unbiased stereology used to identify brain structure borders? Were cell counts averaged per animal? The numbers are expressed in the figures as % of naïve animals, but it is impossible to know what these numbers really mean without more information. Moreover, no micrographs are shown.*

We have added an explanation for c-Fos labeling to the Material and Methods section as follows: “All cell counting experiments were conducted blind to experimental group. The quantification of c-Fos-positive cells was performed with ImageJ software (Scion, Frederick, MD). c-Fos immunoreactive cells were counted bilaterally using at least three sections for each area. Sub-regions of the mPFC, amygdala and hippocampus were outlined as a region of interest (ROI) according to the Paxinos and Franklin atlas [...] These normalized data were expressed as a percentage, and these percentages were averaged across mice in order to produce the mean of each group.” Also we added representative images of c-Fos labeling in the IL, ITC and CeM in Figure 1.

*2) Additionally, it would be useful to know how successful the retrograde tracer was in labeling dopaminergic projections to the IL. Setting aside the question of c-Fos activation, what % of CTB*^*+*^
*cells were also TH*^*+*^*?*

We calculated the percentage of TH^+^ cells in CTB^+^ cells (59.1±4.5%), which has been explained in the Results. Although we agree that it would be useful to know how successful the retrograde trace was labeling the projections, we did not provide it. The percentage of CTB^+^ cells within TH^+^ cells does not tell us how successful, because we do not know exactly what percentage of dopaminergic VTA cells projects to the IL.